# Ex Vivo Evaluation of Mucosal Responses to Vaccination with ALVAC and AIDSVAX of Non-Human Primates

**DOI:** 10.3390/vaccines10020187

**Published:** 2022-01-25

**Authors:** Carolina Herrera, Ronald Veazey, Melissa M. Lemke, Kelly Arnold, Jerome H. Kim, Robin J. Shattock

**Affiliations:** 1Department of Medicine, Imperial College London, London W2 1PG, UK; r.shattock@imperial.ac.uk; 2Tulane National Primate Research Center, School of Medicine, Tulane University, Covington, GA 70433, USA; rveazey@tulane.edu; 3Department of Biomedical Engineering, University of Michigan, Ann Arbor, MI 48109, USA; lemkem@umich.edu (M.M.L.); kbarnold@umich.edu (K.A.); 4US Military HIV Research Program, Walter Reed Army Institute of Research, Silver Spring, MA 20817, USA; Jerome.Kim@ivi.int

**Keywords:** vaccine, mucosal immunology, HIV-1

## Abstract

Non-human primates (NHPs) remain the most relevant challenge model for the evaluation of HIV vaccine candidates; however, discrepancies with clinical trial results have emphasized the need to further refine the NHP model. Furthermore, classical evaluation of vaccine candidates is based on endpoints measured systemically. We assessed the mucosal responses elicited upon vaccination with ALVAC and AIDSVAX using ex vivo Rhesus macaque mucosal tissue explant models. Following booster immunization with ALVAC/AIDSVAX, anti-gp120 HIV-1_CM244_-specific IgG and IgA were detected in culture supernatant cervicovaginal and colorectal tissue explants, as well as systemically. Despite protection from ex vivo viral challenge, no neutralization was observed with tissue explant culture supernatants. Priming with ALVAC induced distinct cytokine profiles in cervical and rectal tissue. However, ALVAC/AIDSVAX boosts resulted in similar modulations in both mucosal tissues with a statistically significant decrease in cytokines linked to inflammatory responses and lymphocyte differentiation. With ALVAC/AIDSVAX boosts, significant correlations were observed between cytokine levels and specific IgA in cervical explants and specific IgG and IgA in rectal tissue. The cytokine secretome revealed differences between vaccination with ALVAC and ALVAC/AIDSVAX not previously observed in mucosal tissues and distinct from the systemic response, which could represent a biosignature of the vaccine combination.

## 1. Introduction

Following disappointing HIV vaccine trials, such as the STEP trial [1], the RV144 trial evaluating the canarypox vector vaccine (ALVAC-HIV [vCP1521]) plus the gp120 AIDSVAX B/E vaccine with a prime and combination boost regime was the first trial showing partial efficacy [2]. However, the HVTN 702 trial in South Africa, modelled on RV144, using ALVAC-HIV and MF59-adjuvanted bivalent subtype C gp120 was halted for non-efficacy with no safety concerns [3] and became the sixth failed completed HIV vaccine trial. Hence, vaccine candidates aiming at eliciting neutralizing antibodies (NAbs) or based on T-cell responses have not yet provided adequate long-term protection. This has led to alternative antigen design and delivery approaches including, among others, sequential multi-immunogen strategies with germline targeting [4,5]; B cell lineage-based vaccine design [6]; formulations for sustained priming of germinal centers (such as osmotic pumps [7], nanomaterials [8], or microneedle patches [9,10]); the use of cytomegalovirus as viral vectors, which have shown long-term efficacy in macaques [11,12]; and vaccine and non-vaccine pre-exposure prophylaxis combinatorial regimes, currently tested in the PrEPVacc trial (ClinicalTrials.gov Identifier: NCT04066881) following proof-of-concept in macaque studies [13,14,15]. Despite mosaic antigen vaccines showing promising results in macaques [16], the HVTN 705/HPX2008/Imbokodo study (ClinicalTrials.gov Identifier: NCT03060629) testing a mosaic-based vaccine in young women at high risk of acquiring HIV was also discontinued due to futility, as announced by the International AIDS Vaccine Initiative (https://www.iavi.org/news-resources/features/iavi-statement-on-results-from-phase-iib-imbokodo-hiv-vaccine-clinical-trial) (accessed on 20 January 2022). The ongoing parallel Phase III HVTN 706/HPX3002/Mosaico trial (ClinicalTrials.gov Identifier: NCT03964415) is testing the safety and efficacy of this vaccine concept in men who have sex with men and transgender individuals. Passive immunization with the broadly NAb (bNAb) VRC01 fully protected macaques against mucosal challenge [17]. However, no efficacy was observed in the AMP trials (HVTN703 and HVTN704) and in vitro neutralization was only observed with the VRC01-sensitive strains isolated among the participants [18,19].

However, RV144 and non-efficacious trials have highlighted the need to develop models that will facilitate comparison between NHPs and humans, thereby increasing the predictive capacity of NHP studies and reducing the late-stage failures of vaccine candidates [20]. Comparative studies between humans and NHPs showed that Rhesus macaques were immunologically more responsive than humans [21]. However, these studies are generally conducted at the systemic level and do not evaluate the vaccine responses in the mucosal portals of viral entry.

The timeline of early Ab responses to HIV-1 has been characterized in plasma from day 8 post-detection of virus in blood [22]. In mucosal secretions, initial IgA responses can be measured within the first 3 weeks. However, the Ab titers required to confer protection have not been determined [23,24], and whether these protective titers will be the same in serum and mucosal surfaces is not known. In addition to delayed B-cell responses, HIV-1 [25,26] and SIV [27,28,29] can induce severe damage to the mucosal B-cell population within the first 80 days of infection. 

Hence, in this NHP study we characterized the Ab responses elicited by vaccination with ALVAC and AIDSVAX, following the RV144 regime, in the mucosal portals of viral entry and compared them with those induced systemically. Furthermore, we analyzed the potential effect of each vaccine on the mucosal and systemic cytokine profile. 

## 2. Materials and Methods

### 2.1. Study Design and Approval

The study was approved by the Tulane National Primate Research Center (TNPRC) Institutional Animal Care and Use Committee (IACUC). TNPRC is accredited by the Association for Assessment and Accreditation of Laboratory Animal Care (AAALAC no. 000594). The TNPRC Office of Laboratory Animal Welfare (OLAW) assurance number is A4499-01 and the U.S. Department of Agriculture registration number is 72-R-0002. Six female Rhesus macaques (Macaca mulatta) of different ages (AG15: 14 years, AK14: 14 years, AR13: 14 years, BG47: 13 years, BG82: 13 years, EJ76: 8 years) were included in the study. The animals were not treated with Depo-Provera. 

In this proof-of-concept study, each animal was its own control with samples taken at baseline 4 weeks prior to vaccination (Figure 1a). The macaques were vaccinated intra-muscularly at day 0 and weeks 4, 12, and 24 with ALVAC-HIV (vCP1521) (1 mL at 10^6.5^ TCID_50_/mL, kindly provided by Sanofi Pasteur), and boosting with the AIDSVAX B/E vaccine (1 mL with 300 μg/mL of MN rgp120 and 300 μg/mL of A244 rgp120, kindly donated by Global Solutions for Infectious Diseases) was administered at weeks 12 and 24 [2].

Mucosal pinch biopsies (10 rectal and 4 cervical) were collected 4 weeks prior to study initiation (baseline) and 2 weeks after each vaccination. At week 26, the animals were euthanized, and cervical, vaginal, rectal, and sigmoidal tissue were resected. Thee rectal and vaginal secretions were collected at the same time points with Weck-Cel sponges, as described previously [30]. Serum was obtained from the blood extractions at baseline (week -4) and 1 and 2 weeks after each vaccination.

Macaques were humanely euthanized with ketamine hydrochloride (10 mg/kg) and tiletimine/zolazepan (Telazol, 8 mg/kg), in accordance with the American Veterinary Medical Association Guidelines on Euthanasia, 2013. 

### 2.2. Cell, Tissue Explant, and Virus Culture Conditions

All cell and tissue explant cultures were maintained at 37 °C in an atmosphere containing 5% CO_2_. TZM-bl cells [31,32,33] were grown in Dulbecco’s minimal essential medium (DMEM) (Sigma-Aldrich, Inc., St. Louis, MO, USA) containing 10% fetal calf serum (FCS), 2 mM L-glutamine and antibiotics (100 U of penicillin/mL and 100 µg of streptomycin/mL). The cells were tested for mycoplasma contamination and confirmed to be mycoplasma-free.

The mucosal tissue specimens were transported to the laboratory and processed less than 1 h after resection. Upon arrival in the laboratory, biopsies and resected tissues were cut into 2–3 mm^3^ explants comprising epithelial and stromal layers for ecto-cervical and vaginal tissue or epithelium and muscularis mucosae for colorectal tissue [34,35]. The tissue explants were maintained with DMEM containing 10% fetal calf serum, 2 mM L-glutamine and antibiotics (100 U of penicillin/mL, 100 µg of streptomycin/mL, and 80 µg of gentamicin/mL). The tissue explants were cultured for 15 days and maintained by harvesting approximately two-thirds of the culture supernatant at days 3, 7, 11, and 15 and refeeding the cultures with fresh medium.

Viral stocks of the HIV-1 circulating recombinant forms CRF01_AE, CM244, and NP1525 [36], provided by the NIH AIDS Research & Reference Reagent Program (http://www.aidsreagent.org) (accessed on 22 January 2022), and SHIV_BaL_ [37,38] were generated by passaging for 11 days through activated PBMCs [39]. 

### 2.3. IgG/A Semi-Quantitative ELISA

Total and anti-gp120 HIV-1**_CM244_**-specific IgG and IgA levels were measured with Rhesus-macaque-specific in-house semi-quantitative ELISAs, as described previously [10,40,41]. Briefly, for the quantification of total Ig medium-binding 96-well ELISA plates (Greiner Bio-one, Kremsmünster, Austria) were coated overnight at 4 °C with 100 μL/well of goat α-human Kappa and goat α-human Lambda (SouthernBiotech, Birmingham, AL, USA) captures antibodies (diluted 1:1200 for IgG or 1:1000 for IgA) at a 1:1 ratio. The plates were washed 4 times with 0.05% PBS and Tween 20 and then blocked with assay buffer (10% DPBS, 0.05% FCS, and Tween 20) for 1 h at 37 °C. Samples diluted in assay buffer were added in triplicate, as well as standards. IgG standard (SouthernBiotech) ranged between 1 µg/mL and 0.0001 µg/mL, and IgA standard (Sigma, St. Louis, MO, USA) ranged between 5 µg/mL and 0.0003 µg/mL. The plates were incubated for 1 h at 37 °C, washed, and detection Ab (goat α-monkey IgG-HRP (AbD Serotec, Kidlington, UK) or goat α-monkey IgA-HRP (Nordic Immunology, Susteren, The Netherlands) diluted 1:5000) was added. Normal, pre-immune monkey control serum (Alpha Diagnostics International, San Antonio, TX, USA) was used as a negative Control The plates were developed with TMB reagent (KPL, for 5 or 10 min to detect IgG or IgA, respectively, and absorbance was read.

For the quantification of anti-gp120 HIV-1**_CM244_**-specific Ig, standard curve wells were coated with α-human Kappa and α-human Lambda, and sample wells were coated with 2.5 µg/mL A244 gp120 D11 monomer (kindly donated by Prof. Haynes, Duke University).

### 2.4. Neutralization

The samples were incubated in serial dilutions with a standardized amount of HIV-1_CM244_ or HIV-1_NP1525_ for 1 h at 37 °C prior to addition to TZM-bl cells. The extent of viral replication was determined by luciferase quantification of cell lysates (Promega, Madison, WI, USA) [34].

### 2.5. Infectivity Assay

The tissue explants were challenged with SHIV_BaL_ for 2 h in a non-polarized system and then washed 4 times with PBS to remove unbound virus. Cervicovaginal and colorectal explants were then transferred onto gelfoam rafts (Welbeck Pharmaceuticals, London, UK). Explants were cultured for 15 days with approximately 50% of the supernatants harvested every 2 to 3 days and replaced with fresh media. The supernatants were used for analysis of the p27 antigen concentration in the culture supernatants at each harvest day by ELISA (SIV p27 ELISA, Zeptometrix Corporation, Buffalo, NY, USA).

### 2.6. Multiplex Cytokine Analysis

The level of twenty-three cytokines (IL-1β, IL-1Ra, IL-2, IL-4, IL-5, IL-6, IL-8, IL-10, IL-12/23(p40), IL-13, IL-15, IL-17, Il-18, sCD40L, G-CSF, GM-CSF, IFN-γ, MCP-1, MIP-1α, MIP-1β, TGF-α, TNF-α, and VEGF) in tissue supernatants were quantified after 24 h of culture by an NHP-specific multiplex bead immunoassay (Merck Millipore, Burlington, MA, USA) using a Luminex 100 System (Bio-Rad, Hercules, CA, USA). 

### 2.7. Statistical Analysis

The cytokine and Ab concentrations were calculated from sigmoid curve fits (Prism v. 9.2.0, GraphPad). All data presented fulfill the criterion of R^2^ > 0.7. The statistical significance of differences between baseline and samples post-vaccination were determined using a non-parametric Kruskal–Wallis test with no correction for multiple comparisons. The responses to vaccination were considered significant when *p* < 0.05. 

Heat maps were completed using Prism with protein levels normalized to the matched explant controls instead of the mean of the control treatments to improve the correction for explant effects and log transformed (base 2). Differentially abundant proteins were analyzed using Ingenuity Pathway Analysis software to determine the biological processes affected by vaccination. The pathways with a minimum of at least two analytes associated and *p* < 0.05 were considered to be enriched. 

Principal component analysis (PCA) models were created in the Eigenvectors PLS toolbox in Matlab using mucosal and systemic cytokine data measured after each vaccination to visualize the variance in the samples based on all of the measured analytes. Every cytokine was assigned a loading, the linear combinations of these loadings creates a principal component (PC). Loadings and PCs were calculated to describe the maximum amount of variance in the data. Each sample was then scored and plotted using their individual response measurements expressed through the PCs. The percent of variance described by each PC is a measure of the amount of variance in cytokine response explained by that respective PC. All data was normalized vs. the baseline data at week -4. Each sample was plotted on the first two PCs. The distinguishing signatures were visualized using an unsupervised hierarchical clustering of the variable importance projection (VIP)-selected cytokines. 

A correlation analysis was performed in R and the correlation matrices of the correlation coefficients between the variables were constructed using Pearson correlation tests with two-tailed *p* values.

## 3. Results

### 3.1. Specific Anti-gp120 HIV-1_CM244_ IgG and IgA Were Detected in Mucosal Compartments following ALVAC/AIDSVAX Vaccination

IgG is known to be the predominant Ig class in secretions from human female genital tract [42,43,44] and IgA has been found to be prevalent in gastrointestinal secretions [45]. To assess Ab levels in the female genital tract and in the colorectum during the RV144 prime-boost immunization regime, we quantified the total and antigen (Ag)-specific IgG and IgA at baseline and two weeks after each vaccination (Appendix A). As expected, the specific anti-gp120 HIV-1_CM244_ Ig levels of both classes were only detected, above baseline (week -4), in the culture supernatants of mucosal tissue explants obtained at weeks 14 and 26, i.e., following the addition of gp120 Ag during the boosting vaccination with ALVAC/AIDSVAX (Figure 1b,c). In cervical explants, this increase was statistically significant at week 26 (2 weeks after the second boost) (*p* = 0.0021 for IgG and *p* = 0.0024 for IgA) and levels (0.55 ± 0.21 μg/mL of IgG and 0.10 ± 0.02 μg/mL of IgA) were similar to those measured in vaginal explants, with higher concentrations of IgG than IgA (0.39 ± 0.19 μg/mL of IgG and 0.07 ± 0.01 μg/mL of IgA). Lower Ab levels were detected in rectal and sigmoidal tissue explants (IgG: 0.11 ± 0.04 μg/mL in rectum and 0.07 ± 0.03 μg/mL in colon; IgA: 0.04 ± 0.003 μg/mL in rectum and 0.04 ± 0.01 μg/mL in colon) than in cervicovaginal tissues (Figure 1d,e) at week 26. Interestingly, contrary to the boost-dependent increase of Ag-specific IgG and IgA in the female genital tract, at week 14 Ab levels in rectal explants were increased (0.07 ± 0.03 μg/mL of IgG and 0.04 ± 0.0035 μg/mL of IgA) to levels similar to those measured at week 26. Higher total Ab levels were also measured in both mucosal tissues following boosting vaccination (Appendix A).

In mucosal secretions and in parallel to tissue explants greater concentrations of IgG and IgA were measured in vaginal fluids than in rectal secretions. However, Ag-specific IgG and IgA in vaginal secretions were significantly increased at week 14 compared to baseline (*p* = 0.0018) (Figure 1f,g), and IgG levels at week 26 were approximately 5 times lower than those measured in cervical and vaginal explant culture supernatants. In rectal secretions, Ag-specific IgG was only found at week 14 at significant levels above baseline (*p* = 0.0116), and no specific anti-gp120 IgA was detected (Figure 1h,i). 

To establish comparisons between the systemic and mucosal compartments, the total and specific anti-gp120 HIV-1_CM244_ IgG and IgA were measured in serum. Both Ag-specific IgG and IgA were significantly increased one week after the first booster (*p* = 0.0277 for IgG and *p* = 0.0277 for IgA) and remained significantly above baseline levels until week 26 (*p* = 0.0037 for IgG and *p* < 0.0001 for IgA) (Figure 1j,k). 

A statistically significant increase in Ab-specific activity, calculated as the ratio of Ag-specific/total Ig, was only observed in female genital tissue explants and in serum (Appendix A). 

### 3.2. Ex Vivo Efficacy of Abs Elicited by Prime-Boosting Vaccination with ALVAC/AIDSVAX

We then evaluated the functionality of the Abs detected mucosally and systemically following boosting, by measuring their neutralization potency against 2 neutralization-resistant/tier 2 isolates [46], the homologous isolate HIV-1_CM244_ and the heterologous HIV-1_NP1525_. Compared to the positive mucosal controls of neutralization (Supplementary Material and Methods; Appendix A) and the baseline activity of tissue culture supernatants and secretions at week -4 prior to vaccination (Appendix A), Abs detected after the last vaccination at week 26 did not significantly neutralize HIV-1_CM244_ (Appendix A). Within the range of dilutions tested, a lower neutralizing activity was observed against the heterologous strain HIV-1_NP1525_ than against the homologous isolate. The neutralization curves with serum revealed a slightly greater potency; however, 100% neutralization was not reached with the lowest dilution tested (dilution 1/4) and 50% inhibitory dilutions ranged between 1/27 and 1/6 against HIV-1_CM244_ and 1/21 and 1/5 against HIV-1_NP1525_ (Appendix A).

Despite the lack of neutralization potency, protection of cervical and rectal explants from all 6 macaques was observed against ex vivo challenge with SHIV_BaL_ (76.2 ± 9.9% of inhibition in cervical explants and 82.3 ± 11.1% in rectal tissue), as shown by a significant decrease in p27 concentrations in culture supernatants from explants obtained at week 26 compared to baseline explants from week -4 (*p* = 0.0094 in cervical explants; *p* = 0.0005 in rectal explants) (Figure 2). 

### 3.3. Proteomic Analysis of Responses to ALVAC and AIDSVAX

To assess if potential modulation of the mucosal environment at the proteomic level could be linked to the observed ex vivo protection, we compared the cytokine/chemokine levels after each vaccination with those found in each animal at the pre-vaccination baseline, week -4. Distinct changes in mucosal cytokine/chemokine levels were observed after each prime and boost vaccination (Figure 3). Two weeks after the first ALVAC vaccination, a downregulated cytokine profile was detected in cervical explants, in contrast to rectal explants (Figure 3a,c). The expression of chemokines (MCP-1 and IL-8) and inflammatory (IL-6 and IL-12) and adaptive (IL-15) cytokines was significantly lower compared to baseline in cervical tissue; however, in rectal explants, chemokines (MCP-1 and IL-8) and anti-inflammatory (IL-10) and angiogenic (VEGF) cytokines were upregulated (Appendix A). The second immunization with ALVAC induced opposite changes in both mucosal compartments. The upregulated cervical profile, with a significant increase of VEGF, contrasted with the downregulation of rectal chemokines (MIP-1β and MIP-1α) and inflammatory (IL-1 and TNF-α) and adaptive (IL-4, IL-13, IL-15, IL-18, IFN-γ, and sCD40L) cytokines (Appendix A; Appendix A). Interestingly, the addition of AIDSVAX in the week 12 boosting induced a downregulated profile in both mucosal compartments 2 weeks after vaccination (week 14), with a significantly decreased secretion of adaptive (IL-2, IL-13, IL-15, and IL-17), inflammatory (IL-12), and anti-inflammatory (IL-1RA) cytokines, chemokine MIP-1β, and antimicrobial protein TGF-α (Appendix A). A functional pathway analysis associated with this anti-inflammatory profile revealed, in both mucosal compartments, downregulated nodes around IL-1α, IL-1β, IL-3, IL-6, IL-17A, IL-33, TNF, and GM-CSF(CSF2) (Appendix A) linked to differential regulation of cytokine production in macrophages and T helper cells by IL-17A and IL-17F, a role as pattern recognition receptors in the recognition of bacteria and viruses, HMGB1 signaling, wound healing signaling, erythropoietin signaling, the Th1 pathway, dendritic cell maturation, and macrophage stimulation, among other canonical pathways. However, similar to the effect of the second ALVAC vaccination, at week 26, after the second boost with ALVAC/AIDSVAX, an upregulated profile was induced in both mucosal compartments (Appendix A). In cervical tissue, two nodes of biological networks centered around the inflammatory cytokine IL-1β and the antimicrobial protein GM-CSF/CSF2 were linked to the stimulation of myeloid cells (Figure 3b). In rectal explants, nodes around IL-6 and IFN-γ, both linked to cell maturation and the stimulation of B cells, were identified among others (Figure 3d).

An analysis of the serum collected 1 and 2 weeks after each immunization revealed a significantly increased expression of the inflammatory cytokine sCD40L at each time point after all four vaccinations (Figure 3e; Appendix A–p). At week 6, functional pathways, including IL-17 signaling, the stimulation of immune cells, and crosstalk between dendritic and natural killer cells, were found to be associated with nodes around CD40L, IL-1β, IL-6, IL-17, and TNF-α, among other nodes (Appendix A). The proteomic profile observed in the mucosal tissues two weeks after the first ALVAC/AIDSVAX (week 14) were not recapitulated systemically. In contrast to the mucosal profiles (Appendix A), upregulated nodes around TNF-α, IL-1β, IL-17, and IFN-γ, among others, were observed in serum (Appendix A). At week 26, analysis of the biological networks associated with vaccination showed small functional nodes around inflammatory (TNF-α, IL-1β, and IL-6) and adaptive cytokines (IL-2, IL-15, and IFN-γ) (Figure 3f), all linked to the stimulation of antigen-presenting cells.

A PCA performed on cytokine levels at week 26 identified 2 PCs that captured 65.8% of the variance and showed defined separation between the cytokines modulated in the rectal, cervical, and systemic compartments (Appendix A). The loading plots revealed cytokines signatures that may be influencing the separation between the levels of rectal, cervical, and serum cytokines along the two PCs (Appendix A). The top three cytokines influencing the first PC were TNF-α, IL-1RA, and IFN-γ, and the top three associated with the second PC were IL-13, IL-6, and MCP-1. 

We then analyzed if separation and biosignatures could be found between the cytokine responses induced after the two ALVAC vaccinations (week 6) and after completion of vaccination with ALVAC/AIDSVAX combination (week 26). A PCA performed with cytokine levels from each compartment at both time points revealed a good separation in the two mucosal compartments but not in the serum (Appendix A). Furthermore, the cytokine signatures influencing the two PCs were distinct in each compartment (Appendix A). In cervical explants, TNF-α, IL-1β, and VEGF influenced the first PC, which accounted for 38.1% of total variance. In rectal explants, IL-15, IL-6, and MCP-1 influenced PC1, representing 44.5% of the total variance. When overlapping all tissues (cervical, rectal, and blood) a good separation between week 6 and week 26 could still be observed, and the majority of week 26-associated cytokines were rectal and cervical (Appendix A).

### 3.4. Correlation Analysis of Responses

We then determined if the cytokine profiles induced by vaccination with ALVAC or the ALVAC/AIDSVAX combination could be correlated with the observed B-cell responses in each compartment. As expected, after vaccination with ALVAC/AIDSVAX, at week 26, different patterns of correlation between the secreted cytokines and the Ag-specific IgG and IgA levels were observed in the cervix, rectum, and serum (Figure 4). In mucosal explants, correlations were observed with inflammatory cytokines and chemokines (Figure 4a,b and Appendix A). The chemokine MCP-1, in cervical cultures, significantly correlated with secreted IgG and IgA. In rectal tissue, Ag-specific IgG levels correlated with IL-1β and MIP-1α and inversely correlated with IL-6 secretion; IgA directly correlated with IL-18 and inversely with IL-5. However, in serum, anti-gp120 HIV-1_CM244_ IgG inversely correlated with the adaptive cytokines IL-15 and IFN-γ (Figure 4c and Appendix A). 

Significant correlations were also found between the cytokine/chemokine changes observed after the two ALVAC vaccinations at week 6 and the specific anti-gp120 HIV-1_CM244_ IgG/IgA levels elicited at the end of the vaccination regime (week 26) (Figure 4d,e,f and Appendix A). In cervical tissue, significant correlations were found with the antimicrobial protein GM-CSF and MIP-1α. Surprisingly, Ag-specific IgG inversely correlated with IL-4 in rectal explants and Ag-specific IgA inversely correlated with IL-5 in cervical explants. Both are adaptive cytokines linked to a Th2 profile. In serum, no significant correlations were found with Ag-specific humoral responses.

A correlation analysis between cytokines and ex vivo protection against SHIV_BaL_ challenge at week 26 (p27 concentrations at day 15 post-challenge and % of inhibition) (Figure 5 and Appendix A) revealed that adaptive cytokines linked to Th2 responses, IL-13, and IL-5, modulated in rectal compartments at week 26, significantly correlated with infectivity (positive correlation) and % of inhibition (inverse correlation). No significant correlation was observed at that time point in cervical tissue cultures. However, the cervical cytokine profile at week 6 showed significant inverse correlations between the inhibition and modulation of the anti-microbial TGF-α and the anti-inflammatory protein IL-1RA. The modulation of inflammatory cytokine IL-6 in rectal explants at week 6 was significantly inversely correlated with inhibition. Additionally, in rectal cultures, levels of chemokines (IL-8, MCP-1, and MIP-1α), inflammatory cytokines (IL-12 and IL-18), GM-CSF, and VEGF at week 6 tended to inversely correlate with inhibition while TGF-α tended to directly correlate.

## 4. Discussion

The mucosal and systemic environment are known to be different; however, limited research has been conducted to assess the compartment-specific responses during vaccination. With this NHP study, we compared the cervicovaginal, colorectal, and systemic responses elicited by vaccination with ALVAC-HIV/AIDSVAX using a prime-boost regime. In line with the closing window to perform placebo-controlled trials linked to the introduction of effective oral pre-exposure prophylaxis interventions and to comply with the principles of the 3Rs (Replacement, Reduction, and Refinement) of animal research [47], we designed this proof-of-concept study using each animal as its own control with baseline samples taken prior to vaccination instead of having a placebo arm and a non-vaccinated control arm. 

Previous studies by Luo et al. have described the absence of Env-specific B cell clonal lineages in the terminal ileum, indicating that Env-specific Abs in rectal secretions are likely produced systemically and transported to intestinal mucosal sites [48]. Our semi-quantitative measurement of Ab levels allowed us to compare the levels of total and anti-gp120 HIV-1_CM244_-specific IgG and IgA in systemic and mucosal compartments for each animal and after each vaccination during the immunization regime. Ag-specific IgG and IgA were detected after the first and second immunizations with ALVAC/AIDSVAX in serum; however, different profiles were observed in the mucosal tissues with significant increases in IgA in cervicovaginal and rectal tissues at week 26 and with significant increases in IgG at weeks 14 and 26 in rectal explants but only at week 26 in cervicovaginal tissues. Furthermore, the Ab levels in mucosal tissues did not consistently correlate with those found in secretions, and this was more evident in the rectal compartment. This might be linked, among other factors, to the higher number of rectal NKp44^+^IL-17^+^ILCs with the use of alum-adjuvanted vaccines compared to other adjuvants, which are known to regulate mucosal integrity [49]. While we evaluated the levels of specific IgG and IgA, it would be important to analyze other Ig classes and subclasses in future studies. The neutralization potency of elicited Abs was also compartment-specific, with lower and less variable IC_50_ values systemically than mucosally. Hence, Ab levels and functionality in serum did not reflect the B-cell responses found in the mucosal compartments. However, the TZM-bl assay is not fully predictive of the neutralization potency in mucosal tissues [50]. This model might not be able to fully recapitulate the activity of mucus-associated proteins, such as mucins, [51] which inhibit HIV via several mechanisms, including, among others: (i) hindering of the viral particle movement [52]; (ii) fusion inhibitory activity [53]; and (iii) inhibition of HIV trans-infection of CD4+T cells by binding to DC-SIGN [54]. Mucins have also been shown to interact with antibodies, leading to Ig accumulation in the mucus and, potentially, to increased local activity. Furthermore, differential binding of IgG and IgA to cervical and cervicovaginal mucus has been described [55].

The sparse amount of tissue and limited volume of the culture supernatants did not allow us to assess the neutralization profile against a broad panel of isolates nor to evaluate potential Ab-dependent cellular cytotoxicity (ADCC) or other non-neutralizing Fc-receptor-mediated Ab effector functions described for the RV144 trial in the systemic compartment [56,57,58,59,60,61,62]. However, we were able to evaluate the potential modulation of cytokines/chemokines induced by each immunization. Systemic and in vitro studies have shown that ALVAC induces a pro-inflammatory profile, the maturation of dendritic cells, and IFN responses [63,64,65,66]. In this study, ALVAC-HIV induced a systemic inflammatory response in line with the data reported by others in macaques immunized with ALVAC expressing either HIV or SIV genes [20,67]. The impacts of ALVAC on the cervical and rectal cytokine profiles were tissue-specific and distinct from that observed in serum; furthermore, these profiles were affected by each vaccination. Systemically, boosting immunization with ALVAC/AIDSVAX induced a similar cytokine profile to that described in humans [68]. Interestingly, the first ALVAC/AIDSVAX boosting induced, at week 14, an anti-inflammatory milieu in both the cervical and rectal tissues, in contrast to the upregulated mucosal profile observed with the second ALVAC/AIDSVAX immunization. Due to the limited number of biopsies obtained, an ex vivo challenge could not be performed after each vaccination or with more than one isolate to assess the efficacy spectrum of the vaccine. Another limitation is that, at the time of the study, we did not have access to SHIV_MN_ or SHIV_A244_, matching the AIDSVAX gp120s, to perform the ex vivo challenge, and we could not compare the ex vivo challenge results with an in vivo challenge. Hence, future work will be required to evaluate if the downregulation induced by the first ALVAC/AIDSVAX boost has an impact on the susceptibility to infection in vivo. Furthermore, the inverse correlation between the Th2 responses (IL-5 and IL-13) and the ex vivo protection observed in the rectal compartment at week 26 will need to be explored to determine if it constitutes a similar correlate of risk to that observed in RV144, where plasma gp120-specific IgA was associated with the risk of infection [69]. 

The tissue explant model is increasingly being used as a pre-clinical tool to reduce the late-stage failure of HIV prevention candidates [70] and in early clinical trials [71,72,73,74,75,76]. Furthermore, a multi-site study has shown that protocol standardization provides measurement consistency among different laboratories [77]. In this study, despite the lack of normalization to account for potential explant size variability, the use of our standardized protocol and sufficient biological replicates for each condition for each animal provided robust data (Appendix A). Furthermore, this model recapitulates the histological and immunological differences between the intestinal and genital mucosae. The colorectal mucosa has a single-cell columnar epithelium, in contrast to the pluri-stratified squamous epithelium of the lower female genital tract. The intestinal lamina propia contains an abundance of highly activated target cells for HIV infection [78,79,80,81]. Furthermore, the colon and rectum produced higher levels of IgA compared to the lower female genital tract, where IgG is prevalent (Appendix A) [82]. 

This study demonstrates that ALVAC modulates the cytokine/chemokine profile in the mucosal portals of viral entry and these responses can be correlated with the Ab levels elicited by ALVAC/AIDSVAX. Furthermore, boosting with the ALVAC/AIDSVAX combination induced an anti-inflammatory milieu, which could be associated with reduced susceptibility to HIV infection. The systemic responses in NHPs during the vaccination regime were similar to those described in clinical trials; however, the responses observed in serum did not correspond to those measured mucosally using the ex vivo tissue explant model. This proof-of-concept study highlights the surprising impact of parental immunization on mucosal responses. Additional studies are required to comprehensively characterize all aspects of the mucosal responses. An important component would be to compare and contrast the mucosal and systemic cellular responses [69,83]. Hence, this study further supports the need to evaluate the efficacy of vaccine candidates at the mucosal level with relevant tissue models and with the aim to refine pre-clinical prioritization in NHPs. 

## Figures and Tables

**Figure 1 vaccines-10-00187-f001:**
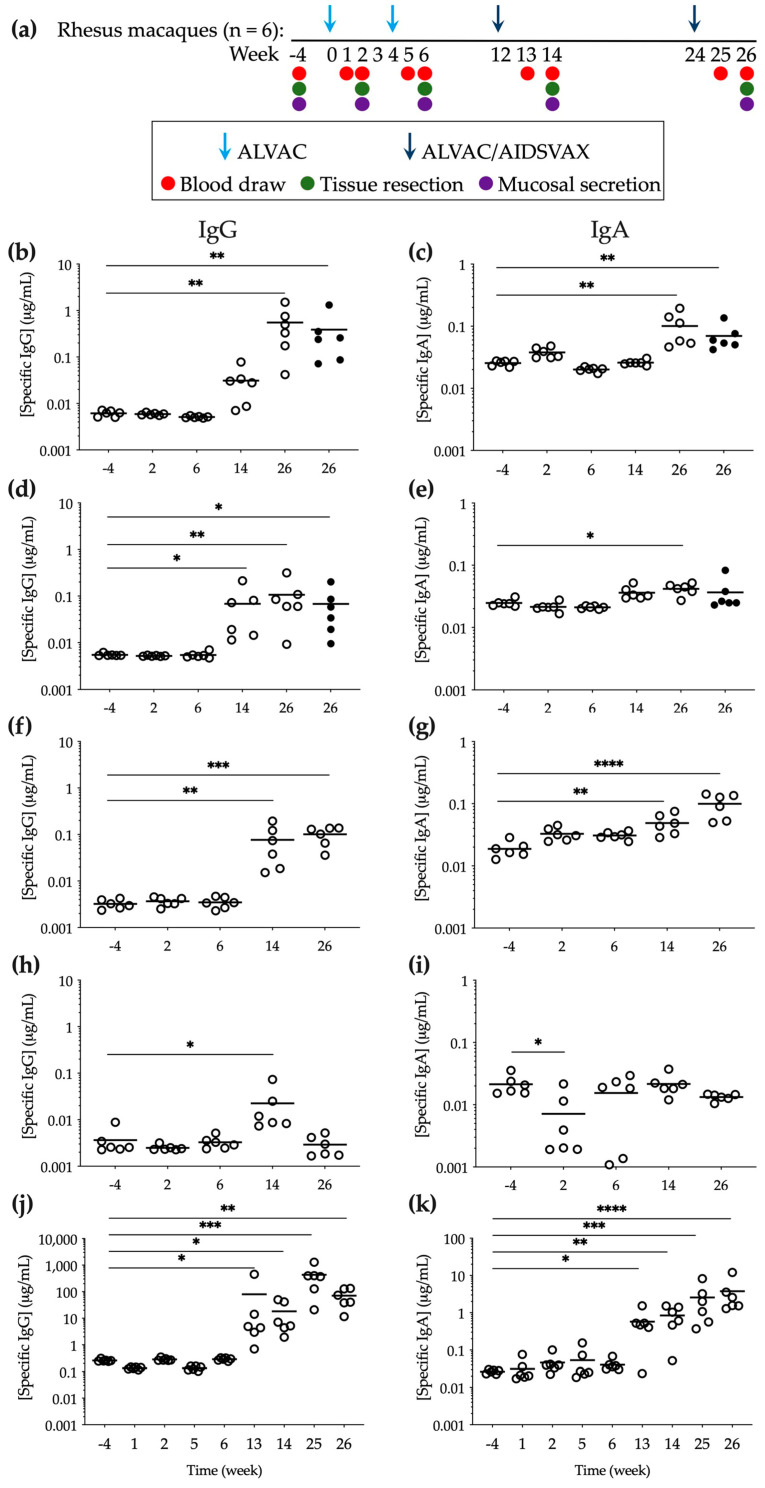
Specific anti-gp120 HIV-1_CM244_ IgG and IgA levels elicited during vaccination with ALVAC/AIDSVAX. (**a**) Immunization and sampling schedule. (**b**–**e**) Explant culture supernatants were harvested during the 15 days of culture of tissues obtained 2 weeks after each vaccination ((**b**,**c**) cervical (open symbols) and vaginal (closed symbols); (**d**,**e**) rectal (open symbols) and sigmoidal (closed symbols)). The data shown are the sum of the Ab concentrations measured in triplicate at harvest days 3, 7, 11, and 15 during the culture period for tissue explants from each macaque. (**f**,**g**) Vaginal and (**h**,**i**) rectal secretions were extracted from Weck-Cel sponges collected from each animal 2 weeks after each vaccination, and Ab concentrations were measured in triplicate. (**j**,**k**) Serum was obtained for each animal from blood collected 1 and 2 weeks after each vaccination and Ab concentrations were quantified in triplicate. Means are shown with lines. Statistical significance: * *p* ≤ 0.05, ** *p* ≤ 0.01, *** *p* ≤ 0.001, **** *p* ≤ 0.0001.

**Figure 2 vaccines-10-00187-f002:**
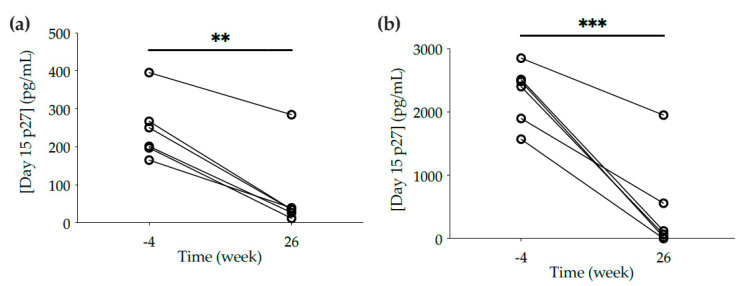
Ex vivo challenge of mucosal explants with SHIV_BaL_. (**a**) Cervical and (**b**) rectal explants cut from biopsies and tissue resected at necropsy were obtained at weeks -4 and 26, respectively. Explants were incubated with SHIV_BaL_ for 2 h, washed four times in PBS, and cultured on gel foam for 15 days. The levels of p27 in the culture supernatants were measured by ELISA at day 15. The data are the means of at least duplicates for each macaque. Lines connect the data from the two time points for each animal. Statistical significance: ** *p* ≤ 0.01, *** *p* ≤ 0.001.

**Figure 3 vaccines-10-00187-f003:**
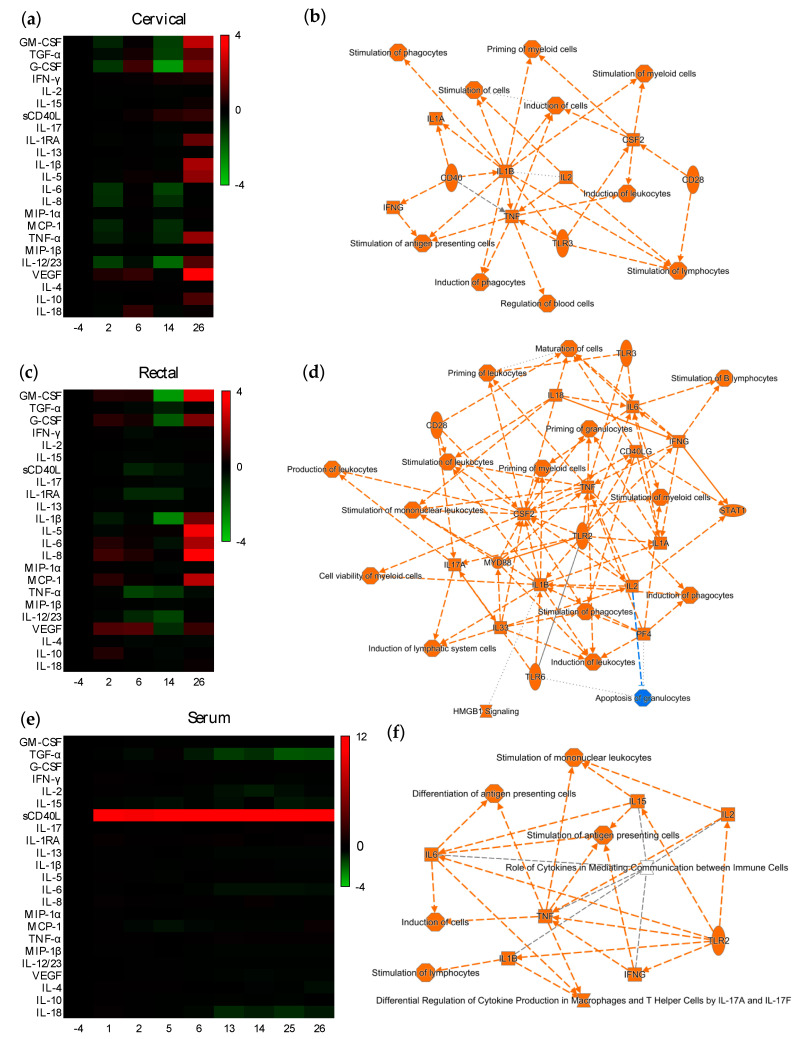
Effect of vaccination on the mucosal and systemic cytokine profile. Heatmap representing cytokines in (**a**) cervical or (**c**) rectal tissues, and in (**e**) serum that are upregulated (red) or downregulated (green) 1 or 2 weeks after each vaccination in comparison to the baseline profiles at week -4. Differences are shown in Log2 from two independent experiments performed in quadruplicate. Interaction maps of the functional pathways associated with the cytokine/chemokine nodes modulated at week 26 in (**b**) cervical explants, (**d**) rectal tissue or (**f**) serum. Colors: blue: downregulated; orange: upregulated. Symbols: oval: transmembrane receptor; square: cytokine/chemokine; diamond: enzyme; inverted triangle: kinase; hourglass: canonical pathway; octagon: function. Lines: solid: direct interaction; dashed: indirect interaction; doted: inferred correlation from Ingenuity Pathway Analysis machine-based learning.

**Figure 4 vaccines-10-00187-f004:**
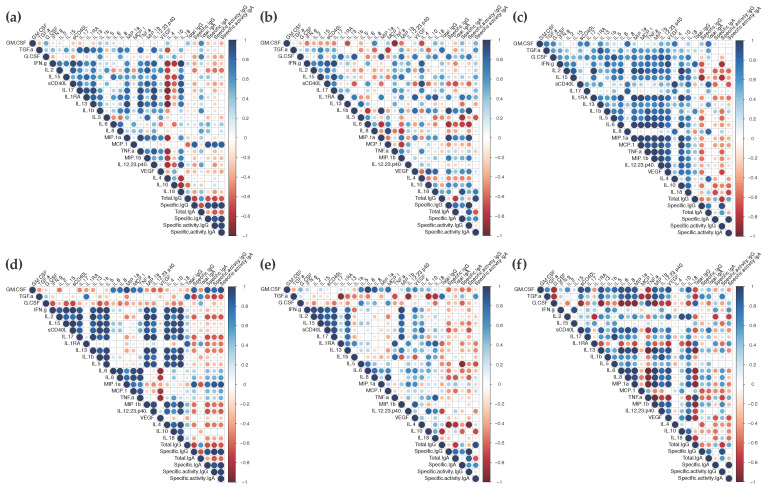
Correlogram of the vaccine responses elicited in the mucosal and systemic compartments. The correlations between the levels of secreted cytokines and the concentrations of total, gp120-specific and the specific activity of IgG and IgA in cervical tissue (**a**,**d**), rectal explants (**b**,**e**), and serum (**c**,**f**) at week 26 (**a**–**c**) or week 6 (**d**–**f**) were analyzed by a correlation matrix. Positive Pearson correlation coefficients are displayed in blue and negative correlations are displayed in red. The color intensity indicated in the side bar is proportional to the correlation coefficient. The size of the circle is proportional to the statistical significance of the correlation. Statistical significance from *p* ≤ 0.05.

**Figure 5 vaccines-10-00187-f005:**
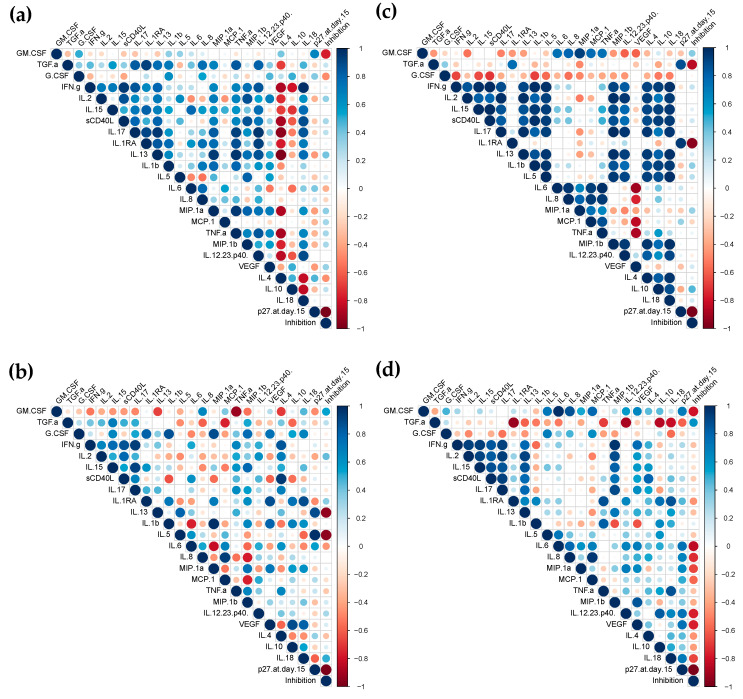
Correlogram of the vaccine responses elicited in mucosal explants. The correlations between the levels of secreted cytokines and susceptibility to ex vivo challenge with SHIV_BaL_ ([p27] (pg/mL) at day 15 and % of inhibition) in cervical (**a**,**c**) and rectal explants (**b**,**d**) at week 26 (**a**,**b**) or week 6 (**c**,**d**) were analyzed by a correlation matrix. Positive Pearson correlation coefficients are displayed in blue and negative correlations are displayed in red. The color intensity indicated in the side bar is proportional to the correlation coefficient. The size of the circle is proportional to the statistical significance of the correlation. Statistical significance from *p* ≤ 0.05.

## Data Availability

Data available upon request.

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
