# Peer review of "Ex Vivo Evaluation of Mucosal Responses to Vaccination with ALVAC and AIDSVAX of Non-Human Primates"

_vaccines, 2022, doi:10.3390/vaccines10020187_

Round 1

Reviewer 1 Report

The manuscript is interesting.

It would be easier to understand if the p values are represented by stars in the figures.

The levels of IgGs and IgAs before vaccination (figure 1) is higher than 0. What is the reason for that? How were the concentrations calculated with the semi quantitative ELISA and what was the minimum concentration that could be detected with the ELISA?

What was the control used in the neutralization assays to make sure that the lack of effect is due to the antibodies not other factors?

What are the p values in figure 2?!

What was the reason to use the Pearson (1-tailed) test rather than Spearman (2-tailed)?

Can the authors provide possible reasons for the differences observed between the cervical and rectal areas?

Author Response

Response to Reviewer 1 Comments

It would be easier to understand if the p values are represented by stars in the figures.

There seems to be an issue with the submission platform because stars have been used in the figures as described in the figure legends. We have mentioned this, and the fact that another reviewer could not see Figure 3 in the text, to the Editor. We apologize about this, and we agree that stars make the figure easier to understand from a statistical point of view.

The levels of IgGs and IgAs before vaccination (figure 1) is higher than 0. What is the reason for that? How were the concentrations calculated with the semi quantitative ELISA and what was the minimum concentration that could be detected with the ELISA?

The concentrations were calculated using the standard curves set up with IgG (ranging between 1 µg/ml and 0.0001 µg/ml) and IgA standard (ranging between 5 µg/ml and 0.0003 µg/ml). Hence, the lower limit of quantification is specific to each Ig. Antibody levels were measured in culture supernatants at different time points (days 3, 7, 11 and 15) during the tissue culture period. To simplify the figures, we show the average sum of the antibody concentrations detected at the four harvest days mentioned above for each macaque. Therefore, addition of baseline levels at each harvest time points for samples collected 4 weeks prior to vaccination, give greater levels than the individual baseline. We have clarified these aspects in the methods section (lines 139-140) and in Figures 1 and S1 legends.

What was the control used in the neutralization assays to make sure that the lack of effect is due to the antibodies not other factors?

We ran two sets of controls: i) we tested the activity of tissue culture supernatants and mucosal secretions obtained at week -4 prior to vaccination as negative controls (new Figure S4); and ii) as positive controls we measured the neutralization activity of culture supernatants from colorectal tissue spiked with different neutralizing Abs (VRC01, 2G12, PGT151 or b6) (new Figure S3). We did not have access to unvaccinated macaque tissue, so we had to use human colorectal tissue obtained from surgical resections as described in the new Supplementary Material and Methods. We agree that these data improve the quality of the manuscript and we have modified the section of neutralization results (lines 265-267).

Multiple factors could affect the neutralization potency. We have previously described that the TZM-bl assay is not fully predictive of the neutralization potency in mucosal tissues (Cheeseman H. et al. 2017). This model might not be able to recapitulate the inhibitory activity of mucus-associated proteins in tissue samples, such as mucins. We have added a section in the discussion about this point (lines 493-501).

What are the p values in figure 2?!

We agree that the statistical analysis was missing, and we have now included it in the text and with stars in the figure.

What was the reason to use the Pearson (1-tailed) test rather than Spearman (2-tailed)?

We used a Pearson correlation assuming a Gaussian distribution of values and aiming to detect only lineal relationships. However, we chose two-tailed p values for the statistical analysis, which has now been clarified in the text (lines 196-197).

Can the authors provide possible reasons for the differences observed between the cervical and rectal areas?

Thank you for this question, discussing the differences was indeed missing in the discussion. We have summarized the differences between both compartments in lines 533-541.

Reviewer 2 Report

This manuscript mainly reported to assess the ex vivo mucosal responses in NHP animal model after ALVAC/AIDSVAX Vaccination (to mimic the RV144 clinical trial). There are numerous major and minor concerns regarding this manuscript

1, Although the RV144 was the was the first trial showing partial protection against HIV-1 infections, the similar ALVAC/AIDSVAX strategy called HVTN702 trial had been terminated recently due to no observed protection. Authors should give a comment on it.

2, only 6 monkeys were invovled in this study, and a control group was missed.

3, Figure1 : P value or asterisk symbol is missed in this panel. The font of X/Y axis is strange. Lack of a schedule of vaccination.

4, Figure 2: No Statistical analysis is shown. Ex vivo challenge of mucosal explants with SHIVBaL was performed in this study, and why not conduct a in vivo challenge in this monkey model?

5, Figure 3: no picture is shown!

6, Figure 4 and Figure 5:  It is hard to understand.

Author Response

Response to Reviewer 2 Comments

1, Although the RV144 was the was the first trial showing partial protection against HIV-1 infections, the similar ALVAC/AIDSVAX strategy called HVTN702 trial had been terminated recently due to no observed protection. Authors should give a comment on it.

We have added this trial in the introduction (lines 36- 39).

2, only 6 monkeys were involved in this study, and a control group was missed.

We agree that ideally a placebo control group could have been included and further animals would have increased statistical power; however, with funding limitations the decision was taken for this proof-of-concept study, to use each animal as its own control with samples taken at baseline 4 weeks prior to vaccination. The project was funded by CHAVI/HVTN/NIH with this design to comply with the principles of the 3Rs (Replacement, Reduction and Refinement) to perform more humane animal research (https://nc3rs.org.uk/the-3rs), and in line with the ethical issues of placebo-controlled HIV prevention trials linked to the introduction of effective prevention interventions such as Truvada or F/TAF. We have added a sentence in the ‘Study design and approval’ section (lines 85-86) and in the Discussion (lines 469-474) to explain the study design.

3, Figure1: P value or asterisk symbol is missed in this panel. The font of X/Y axis is strange. Lack of a schedule of vaccination.

Looks like there is a formatting issue because stars have been used in the figures as described in the figure legends. We have mentioned this to the Editor because Figure 3 is also in the text submitted to the journal. We apologize about this, and we agree that stars make the figure easier to understand from a statistical point of view.

We agree that a vaccination schedule will be helpful, and we have added an immunization and sampling schedule in this Figure.

4, Figure 2: No Statistical analysis is shown. Ex vivo challenge of mucosal explants with SHIVBaL was performed in this study, and why not conduct a in vivo challenge in this monkey model?

We agree that the statistical analysis was missing, and we have now included it in the text and with stars in the figure. At the time of evaluation of the grant application, the Funder only approved ex vivo challenge analysis to obtain proof-of-concept of the feasibility of our hypothesis. Hence, funding restrictions forced us to remove the in vivo challenge part of the study to further validate the ex vivo study. We have added this limitation in the discussion (Line 521-522).

5, Figure 3: no picture is shown!

As mentioned above, a technical glitch must have affected the text because this figure was submitted. This issue has been raised with the Editorial Team.

6, Figure 4 and Figure 5:  It is hard to understand.

Thank you for this comment that will improve the quality of the figures. Both figures have been reformatted within the page limits (as well as the matching supplementary figures), however, we could submit separated files as supplementary material of individual components of Figure 4 and 5 with further increased size if necessary.

Reviewer 3 Report

The work is of value to understanding the effects of HIV vaccines. However, there are some significant issues that wold need to be addressed before it can be properly assessed.

Not really mentioned in figures and text had unimmunized animal for comparison or baseline. And the mucosal antibodies are not really well-characterized with controls. On the benefit that these are just left out, major revision is recommended.

Introduction is a bit scant. Can provide more information on the progress of HIV vaccines, etc.

Would suggest mentioning ethics approval at the start of the study design.

If possible, it would have been useful to separate the evaluation of IgA1 and A2 equivalents, and given that it is mucosal, IgE should also be considered. This can be performed with qPCR if ELISA were not set up.

Line 163, it should be noted that IgG is not a subclass but a class IgG1234,etc are subclasses.

Given that explant size may affect the quantification of the cytokines and antibodies, please state explicitly any normalization and control where used.

Is there an animal that is not vaccinated to be used as control for comparison? Please state it explicitly for comparison of the effects and baseline.

There is figure 3 caption but no Figure 3?

Figure 1 fonts should be consistent with text. And can be improved if the axes are the same. Figure 1 can be larger for better viewing especially when printed.

Figures 4 and 5 are impossible to read and see

Author Response

Response to Reviewer 3 Comments

Not really mentioned in figures and text had unimmunized animal for comparison or baseline. And the mucosal antibodies are not really well-characterized with controls. On the benefit that these are just left out, major revision is recommended.

We agree that ideally a placebo control group could have been included; however, budget restrictions by the Funder with approval of a reduced design forced us to remove the placebo control arm. However, we decided to use each animal as its own control which complies with the principles of the 3Rs (Replacement, Reduction and Refinement) to perform more humane animal research (https://nc3rs.org.uk/the-3rs) and with the raising ethical concerns around placebo-controlled trials linked to the introduction of effective drug-based prevention strategies. We have added a sentence in the ‘Study design and approval’ section (lines 85-86) and in the Discussion (lines 469-474) to explain the study design.

When assessing the neutralizing activity of mucosal antibodies, we ran two sets of controls: i) we tested the activity of tissue culture supernatants and mucosal secretions obtained at week -4 prior to vaccination as negative controls (new Figure S4); and ii) as positive controls we measured the neutralization activity of culture supernatants from colorectal tissue spiked with different neutralizing Abs (VRC01, 2G12, PGT151 or b6) (new Figure S3). We did not have access to unvaccinated macaque tissue, so we had to use human colorectal tissue obtained from surgical resections as described in the new Supplementary Material and Methods. We thank the reviewer for this comment which improves the quality of the manuscript and we have modified the section of neutralization results (lines 265-267).

Introduction is a bit scant. Can provide more information on the progress of HIV vaccines, etc.

We thank the reviewer for this comment which has improved the introduction. We have added a section on the progress of HIV vaccines within the context of the manuscript (lines 36-56)

Would suggest mentioning ethics approval at the start of the study design.

We have moved the paragraph describing ethics approval as suggested.

If possible, it would have been useful to separate the evaluation of IgA1 and A2 equivalents, and given that it is mucosal, IgE should also be considered. This can be performed with qPCR if ELISA were not set up.

We agree it would be interesting to further characterize the antibody response; however, due to the limited volume of culture supernatant available in each well we could not perform at the time in-depth isotyping. At the time when this study was performed, we had not introduced in our laboratory multiplexing technologies for antigen-specific Ig characterization. We have added a sentence in the discussion mentioning this limitation (line 489-490).

Line 163, it should be noted that IgG is not a subclass but a class IgG1234,etc are subclasses.

We apologize about this typo which has not been corrected.

Given that explant size may affect the quantification of the cytokines and antibodies, please state explicitly any normalization and control where used.

We agree that normalization would have been useful. At the time when this study was performed, we did not normalize for weight or protein content, and instead for each sample from each animal, we performed biological replicates which showed good reproducibility despite the lack of normalization. We have mentioned this limitation in the discussion (lines 529-533).

Is there an animal that is not vaccinated to be used as control for comparison? Please state it explicitly for comparison of the effects and baseline.

We agree that a non-vaccinated animal would have further confirmed the baseline controls; however, due to funding limitations, for this proof-of-concept study, we used each animal as its own control with samples taken at baseline 4 weeks prior to vaccination. We have added this limitation in the discussion (lines 469-474).

There is figure 3 caption but no Figure 3?

We have mentioned to the Editor the fact that despite the Figure being in the text, some reviewers could not see it. We apologize about this.

Figure 1 fonts should be consistent with text. And can be improved if the axes are the same. Figure 1 can be larger for better viewing especially when printed.

This Figure has also been affected by this technical glitch in the submission system because the font is the same as the text.

Figures 4 and 5 are impossible to read and see

Thank you for this comment that will improve the quality of the figures. Both figures have been reformatted within the page limits, however, we could submit separated files as supplementary material of individual components of Figure 4 and 5 with further increased size if necessary.

Round 2

Reviewer 2 Report

I have read this manuscript entitled “Ex Vivo Evaluation of Mucosal Responses to Vaccination with ALVAC and AIDSVAX of Non-human Primates” again with potential interest. Authors have addressed some concerns, but not all. For example, authors stated that the reason of missing a control group is due to funding limitations. It is understood to do not perform a in vivo experiment with a placebo control or a non-vaccinated control, but I think it should be easy to acquire some naive monkeys’ samples for in vitro assay? Moreover, there are some new concerns:  This study investigated the humoral responses, however, the cellular response at the mucosal tissues/surfaces is also very important in controlling viral infections. There is no any comment or data reported in this manuscript.  

Author Response

I have read this manuscript entitled “Ex Vivo Evaluation of Mucosal Responses to Vaccination with ALVAC and AIDSVAX of Non-human Primates” again with potential interest. Authors have addressed some concerns, but not all. For example, authors stated that the reason of missing a control group is due to funding limitations. It is understood to do not perform a in vivo experiment with a placebo control or a non-vaccinated control, but I think it should be easy to acquire some naive monkeys’ samples for in vitro assay?

Unfortunately, we are not able to meet the reviewers request. It is important to recognize that each macaque serves as its own control. This is seen as acceptable by the other reviewer. Our IACUC would not grant permission to necropsy naïve animals for additional controls. As mentioned in the updated version of the manuscript, we are required to adhere to the 3Rs. This and the timeline required by the editorial team for resubmission mean that that we cannot address this through inclusion of additional experiments. We have referenced the reviewer’s concerns by raising this as a caveat in the discussion section (lines 407-411).

Moreover, there are some new concerns:  This study investigated the humoral responses, however, the cellular response at the mucosal tissues/surfaces is also very important in controlling viral infections. There is no any comment or data reported in this manuscript.  

We agree that it would be interesting to analyze the mucosal cellular responses and compare them to those described systemically in the literature (Haynes et al. 2012; de Souza et al. 2012). This would require additional tissue samples which were not available considering the limited number of biopsies that could be obtained from each animal and the small size of the macaque cervix. Unfortunately, this interesting aspect falls beyond the scope of this current study. We have added a comment in the discussion to reflect this (lines 494-497).

Reviewer 3 Report

While it may be acceptable without a placebo group and this may be addressed by the approval of animal ethics, serious problems still persist in this article.

The lack of normalization of samples based on the explants which can easily confound the expression levels measured must at least be somewhat normalized for meaningful results.

If the claim is mucosal responses, I think it is expected to  have better characterization of the antibody classes and subclasses - it is just a qPCR with primer design etc.

These problems are generally too important to ignore and they all stack up to undermine the outcomes of the work as much as trying to be sympathetic

Smaller problems if the authors wish to work on it, are that Figures 4 and 5 are not easy to work through. Better presentation needs to be performed or thought through.

Author Response

The lack of normalization of samples based on the explants which can easily confound the expression levels measured must at least be somewhat normalized for meaningful results.

While we recognise the reviewer’s concerns we respectfully point out that the biological replicates show a high level of consistency and reproducibility of data among replicates as indicated by the tight error bars obtained in all assays. We have added two supplementary figures (Figure S1 and S3) with the data of IgG/A ELISAs with error bars for each animal and commented on this in the manuscript (lines 477-480).

 If the claim is mucosal responses, I think it is expected to have better characterization of the antibody classes and subclasses - it is just a qPCR with primer design etc.

We agree that it would be interesting to test additional antibody subclasses beyond antigen specific IgA/IgG. It is important to recognise that qPCR would only determine subclass of total and not antigen specific antibody. It might be possible to determine subclass by a multiplex assay, however the volume of sample required to perform this analysis was not available in addition to that required for the total and specific IgG/A ELISAs. This is a proof-of-concept study, and in new studies we will explore how to multiplex the analysis that can be done with 200 µl of culture supernatant. Furthermore, it would not be possible to complete this given that the Editor has given us a few days to resubmit the manuscript and this would involve starting a new study with new macaques. We have made reference to this in the discussion section as a potential caveat (lines 427-428)

These problems are generally too important to ignore and they all stack up to undermine the outcomes of the work as much as trying to be sympathetic.

We acknowledge that these are important considerations for future studies but would like to emphasize that this is a proof-of concept study that opens the door to further investigation and potential funding. The nature of this proof-of-concept study means that it was not possible to comprehensively characterize all aspects of the mucosal responses but rather to highlight our novel findings that parenteral immunization can impact on mucosal responses.  We agree with the reviewer that this novel finding deserves additional future investigation, but this falls beyond the scope of this initial manuscript. We have alluded to this in the discussion (lines 494-497).

Smaller problems if the authors wish to work on it, are that Figures 4 and 5 are not easy to work through. Better presentation needs to be performed or thought through.

We are sorry that the reviewer found these figures difficult to work through, they represent the standard format for presenting correlation data. We have added additional description of the symbols in both figure legends to clarify the correlation matrices (lines 377-379 and lines 399-401). We also refer the reviewer to the presentation provided by new supplementary figures (Figure S1 and S3) where smaller components of the correlation matrices are shown for further clarity.

Round 3

Reviewer 2 Report

I do not think authors have carefully addressed previous concerns.

For example, "We have referenced the reviewer’s concerns by raising this as a caveat in the discussion section (lines 407-411) ", but I don't find the corresponding explanation in line 407-411 to address that concern.

As for "We have added a comment in the discussion to reflect this (lines 494-497)", the line 494-497 is "Acknowledgments" ?

Author Response

There seems to have been a formatting issue again with lines being correct in the version submitted. Thank you

Reviewer 3 Report

I cannot see that the problems are addressed. 

Author Response

Figures have been updated. Once more we would like to emphasize that this is a proof-of concept study that opens the door to further investigation.

Thank you